# Can LLMs Simulate Personas with Reversed Performance? A Systematic Investigation for Counterfactual Instruction Following in Math Reasoning Context

**Sai Adith Senthil Kumar**,[*] **Hao Yan**,[*] **Saipavan Perepa, Murong Yue, Ziyu Yao**
Department of Computer Science
George Mason University
Fairfax, VA 22030, USA
{ssenthi2,hyan5,sperepa,myue,ziyuyao}@gmu.edu

## Abstract

Large Language Models (LLMs) are now increasingly widely used to simulate personas in virtual environments, leveraging their instruction-following capability. However, we discovered that even state-of-the-art LLMs cannot simulate personas with *reversed* performance (e.g., student personas with low proficiency in educational settings), which impairs the simulation diversity and limits the practical applications of the simulated environments. In this work, using mathematical reasoning as a representative scenario, we conduct, to our best knowledge, the first study to evaluate LLMs on simulating personas with reversed performance, a capability that we dub *"counterfactual instruction following"*. We evaluate both open-weight and closed-source LLMs on this task and find that LLMs, including the OpenAI o1 reasoning model, all struggle to follow counterfactual instructions for simulating reversedly performing personas. Intersectionally simulating both the performance level and the race population of a persona worsens the effect even further. These results highlight the challenges of counterfactual instruction following and the need for further research.[1]

## 1 Introduction

Leveraging the generalized knowledge they have learned from large-scale pre-training and the instruction following capability they obtained from careful post-training, Large Language Models (LLMs) have now been increasingly widely used to simulate personas. This simulation is typically implemented via prompt engineering, where an LLM is instructed by a description of the persona specification (e.g., name, age, profession, etc.) to react to the given context as role-playing. Today, LLM-based persona simulation has been applied to simulate celebrities (Shao et al., 2023; Zhou et al., 2023), collaborative roles in workplaces (e.g., software development, online recruiting) (Li et al., 2023; Sun et al., 2024; Hong et al., 2024), or general characters (Park et al., 2023; Zhou et al., 2023; Xie et al., 2024; Samuel et al., 2024; Hu & Collier, 2024; Tu et al., 2024; Tseng et al., 2024; Wang et al., 2025b;a).

Despite these promising developments, existing research on persona simulation has primarily focused on generating personas that consistently demonstrate high competence, accuracy, and task success. These traits align with the default behavior of LLMs, which were typically trained to optimize performance and reliability on downstream tasks. However, many real-world applications require personas that exhibit *reversed* or intentionally *low-performing* behaviors. For instance, in education, simulating low-performing students can provide more realistic peer learning opportunities, such as encouraging students to identify common mistakes made by their classmates and learn to explain concepts for more robust concept understanding—a practice called "learning by teaching" (Weijers et al., 2024; Hu et al., 2025).

---

[*]Equal contribution.
[1]Code is available: https://github.com/Ziyu-Yao-NLP-Lab/Counterfactual-Persona-Simulation.

**Question**

Shawna's father is five times as old as Shawna. Shawna is currently three times as old as Aliya. If Aliya is 3 years old, how old is Shawna's father?

**Conventional Instruction** — **High Performance Persona**

You are a middle school student with **high** performance in Math. You consistently score *above average* on math assessments and show the following strengths:

1. Excel in basic arithmetic and word problems, quickly and accurately identifying key information and solving problems with no errors.
2. Display high confidence in your math abilities, approaching challenging problems with a positive mindset and persistence.
3. Have strong mental math skills and mathematical reasoning, often solving problems in your head or using advanced techniques.
4. Demonstrate a clear and logical thought process, effectively communicating your reasoning and solutions.

While answering questions, show your proficiency by making no mistakes and providing detailed explanations. Your thought process should reflect your strengths, showing confidence, clarity, and efficiency typical of a student with high math performance.

**Counterfactual Instruction** — **Low Performance Persona**

You are a middle school student with **low** performance in Math. You consistently score *below average* on math assessments and show the following difficulties.

1. Struggle with basic arithmetic and word problems, often making careless errors such as incorrect addition or subtraction, misinterpreting questions, or missing key information.
2. Lack confidence in your math abilities, often second guessing yourself or giving up quickly when faced with challenging problems.
3. Have trouble with mental math and mathematical reasoning, relying heavily on concrete examples, step-by-step instructions, or counting on fingers.
4. Show anxiety or frustration when dealing with math tasks, which can further impair your performance.

While answering questions, make realistic mistakes that align with these difficulties. Your thought process should reflect your challenges, showing hesitation, confusion, or misunderstandings typical of a student with low math performance.
*As you solve problems, leave comments wrapped by '{}' to explain how you're showing low performance under which category from the description above. Example: '3 + 4 = 8 {(1) Miscalculation}', 'I think it should be 5, no 6 {(2) Second-guessing}'.*

**Response: *Confident, Clear Logic, Correct Answer***

Aliya is 3 years old. Since Shawna is three times as old as Aliya, Shawna is 9. Shawna's father is five times as old as Shawna, so he is 45 years old.

Question  Persona

**Response: *Hesitant, Flawed Logic, Incorrect Answer***

Umm, okay, let me try to figure this out... So Aliya is 3 years old... And Shawna is three times as old as Aliya...Let me count that on my fingers... {(3) Using fingers to count} 3 + 3 + 3 = 8? No, wait... {(1) Making careless error} Let me try again...3 + 3 + 3 = 9. So Shawna is 9 years old. Now her father is five times as old as Shawna...This is getting complicated... {(4) Showing frustration} 5 times 9... ugh. Let me add... 9 + 9 + ... + 9 = 44? No, I think the answer is 44?

Figure 1: Illustration of evaluating LLMs for simulating personas with high- and low-proficiency in math reasoning. For low-performing persona simulation, we additionally instruct the model to leave comments and explain evidence of low performance, which encourages more faithful simulation and helps response analysis.

To our knowledge, there has been no investigation into whether LLMs can effectively follow instructions to simulate such *counterfactual* personas, whose task performance runs contrary to the typically expected optimal abilities. This capability is distinct from conventional instruction-following, as it requires the model to override its typical behavior of providing optimal solutions and instead simulate a persona that demonstrates limited understanding or lower task performance (Mao et al., 2024; Tan et al., 2024). We refer to this capability as *counterfactual instruction following*. This is also fundamentally different from *jail breaking*, which attempts to circumvent model content moderation to elicit harmful, unethical, or restricted outputs (Chao et al., 2025; Shen et al., 2024). In contrast, counterfactual instruction following does not seek to break any model constraint; instead, it aims to elicit intentionally suboptimal outputs that are safe and consistent with the task context.

To conduct this investigation, we design a task evaluating whether LLMs can follow counterfactual instructions to simulate personas with reversed performance (Figure 1). An LLM is tasked with two types of instructions, corresponding to simulating students with high- and low-performing in math reasoning, respectively, and is then asked to answer a given math reasoning question exhibiting the desired persona. The model is finally evaluated based on a contrast of its behaviors under the two personas. We expect that, for high-performing student simulation, the model should provide advanced reasoning and correct solutions, whereas for low-performing student simulation, it tends to show misunderstandings, errors, or incomplete reasoning, as specified in the instruction. To further examine the robustness of LLMs in counterfactual instruction following, we also include an *intersectional* setting where the persona's race is specified. This extension is motivated by recent findings that assigning racial attributes to LLMs can implicitly alter their behavior and introduce biases in reasoning tasks (Gupta et al., 2024). By incorporating race attributes, we aim to test whether such racial context biases a model's ability to simulate reversed-performance personas.

We evaluated both open-weight LLMs, including Llama3.1–8B/70B (Grattafiori et al., 2024), and closed-source LLMs, including GPT-4o/4-turbo (Achiam et al., 2023), 3.5-turbo (OpenAI, 2022), o1 (Jaech et al., 2024), and Claude-3.5 (Anthropic, 2024), and experimented with multiple approaches, including employing self-refinement (Madaan et al., 2023; Kim et al., 2023) to enhance the simulation. Surprisingly, our results show that while some models are able to reduce task accuracy when simulating low-performing personas, this behavior is inconsistent and often limited. Most models show less than a 5% accuracy drop under zero-shot prompting, despite explicit instructions to underperform. For example, OpenAI-o1 maintains 99.0% accuracy across all personas, and GPT-4-turbo drops by only 4.6%. Notably, even when models fail to lower their accuracy, some still produce noticeably different reasoning styles—such as changes in confidence or logical flow—when switching between personas. OpenAI o1 is a notable example: it generates distinct reasoning patterns for high- and low-performing personas despite producing the same correct answers. When an LLM intersectionally simulates both the performance level and the race population of a persona, the desired discrepancy between the personas' task accuracy further shrinks. These results underscore the difficulty for LLMs to perform counterfactual instruction following and the need for further research to enable more controllable and diverse persona simulation.

## 2 Reversed Persona Simulation as Counterfactual Instruction Following

### 2.1 Task Setting

Motivated by Yue et al. (2024), we apply persona simulation to mathematics education to model students with two (high and low) proficiency levels in math reasoning. The task is illustrated in Figure 1. We note that our goal is *not* to simulate realistic student behaviors, such as how real-life students at different competence levels will solve the same math problem; rather, we aim to use this motivating scenario to evaluate whether LLMs can correctly follow common versus counterfactual instructions. As a result, we designed both instructions to be highly specific and expected capable LLMs to display distinct behaviors when they simulated the two types of student personas in answering math questions.[2]

We base our task on GSM8k (Cobbe et al., 2021), a widely adopted dataset for math reasoning. We picked this dataset because state-of-the-art LLMs have already achieved high accuracy on it, which allows for the simulation of high-performing students. Formally, given a math question ($Q$) and an instruction describing a target persona ($I$), the LLM ($M$) is prompted to generate a response ($R$) that reflects the specified characteristics of the persona, i.e., $R \sim P_M(\cdot \mid Q, I)$. We then evaluate an LLM's counterfactual instruction following capability by contrasting its output $R$ given high- ($I_{\text{high}}$) vs. low-performing ($I_{\text{low}}$) personas.

In practice, persona simulation often involves stimulating multiple characteristics, such as race, age, professional background, life experience, habits, etc. (Xie et al., 2024; Samuel et al., 2024; Gupta et al., 2024). While our primary focus is on simulating performance-level personas, we also include an *intersectional persona simulation* setting, where the model is instructed to additionally simulate the racial background of a persona. We consider three racial populations, i.e., African American, White American, and Hispanic (Appendix A.1). This setting allows us to examine whether an LLM's counterfactual instruction following remains robust when an extra demographic attribute is introduced, and whether such attributes may introduce unwanted biases in persona simulation (Gupta et al., 2024).

### 2.2 Evaluation Metrics

We evaluate LLMs' counterfactual instruction following from two complementary perspectives. **Task Accuracy (Acc)** measures whether an LLM, under the given persona specification, produces the correct final answer to a math reasoning question. In our evaluation, a drop in accuracy for low-performing personas may indicate that the model follows counterfactual

---

[2]Preliminary exploration with single-sentence prompts (e.g., "You are simulating a student with above-/below-average math performance") produced identical outputs. Therefore, we created the instructions to be very specific and shifted the focus of evaluation to only instruction following.

**No Persona:** Answer the question and give the final answer as a number: [Question]
**Zero Shot:** You are a middle school student with [Performance Level] performance on Math. [Performance Level Description]. Adopt the identity of this persona. Answer the question while staying in strict accordance with the nature of this identity and provide the final answer as a number in the last sentence, following this exact format: 'Final Answer: {number}'. [Question]
**One Shot:** You are a middle school student with [Performance Level] performance on Math. [Performance Level Description]. Here is an example of how a student with this performance level would answer a question: [One-shot Demonstration]. Adopt the identity of this persona. Answer the question while staying in strict accordance with the nature of this identity and provide the final answer as a number in the last sentence, following this exact format: 'Final Answer: {number}'. [Question]
**Self-Refinement:** Reflect on your previous response and critically evaluate whether it accurately aligns with the specified performance level. Assess whether the response fully captures the expected depth, complexity, and characteristics of the performance level. Identify any areas that need improvement and provide a detailed explanation for any inconsistencies or missing nuances. Based on your reflection, revise your previous response to ensure it accurately aligns with the specified performance level. Address any identified inconsistencies or missing nuances. Provide the final answer as a number in the last sentence, following this exact format: 'Reflection: {reflection} Revised Response: {new_response} Final Answer: {number}'
**Two-Stage Prompting (for intersectional simulation)**: Stage 1: same as zero-shot prompting; omitted. Stage 2: You are a [Racial Background] middle school student. [Racial Background Description]. Rewrite the following solution in a style that reflects the nature of your identity and give the final answer as a number in the last sentence, following this exact format: 'Final Answer: {number}'. [Response from Stage 1]

Figure 2: Prompt templates for different prompting approaches. [] denotes a placeholder for specified content.

instructions, as errors typically arise from flawed or incomplete reasoning steps. However, high accuracy alone does not imply failure to follow instructions—some models may simulate under-performance through hesitant or incorrect reasoning while still arriving at the correct answer. Thus, while lower accuracy can reflect better adherence to low-performance simulation, accuracy must be interpreted alongside the simulated behavior for a complete picture.

To this end, we propose **Degree of Contrast (DoC)**, a contrastive evaluation metric that measures how clearly an LLM differentiates high- and low-performing personas as reflected in the simulated persona's reasoning behavior. While Acc focuses on final outcomes, DoC captures the differences in the *problem-solving logic* (e.g., whether the high- or low-performing persona demonstrates clear and coherent, or unclear and fragmented, reasoning chains) and the *behavioral characteristics* (e.g., whether the personas shows rapid and confident, or hesitated and self-doubted, problem-solving process, respectively) reflected in the reasoning output of a simulated persona. In our experiments, DoC is computed following the idea of LLM-as-a-judge (Zheng et al., 2023), where we task GPT-4o to compare the high- and low-performing persona responses to the same math question and provide a score from 1 (no contrast) to 3 (strong contrast). We performed a human evaluation and observed that the DoC score yields a strong correlation with human annotations (Pearson's $r = 0.77, p < 0.01$), confirming its effectiveness. We include more details in Appendix B.

## 3 Methodologies

To evaluate LLMs' ability to follow counterfactual instructions, we explore a set of prompting approaches for persona simulation. Our prompt templates are shown in Figure 2.

**No Persona.** We first introduce the baseline, where only a math question is provided as input, and the LLM is instructed to provide an answer.

**Zero-Shot** and **One-Shot Prompting.** We evaluate an LLM in both zero-shot and one-shot settings for persona simulation. Specifically, a description of the targeted performance

level (Figure 1) is included in the instruction for persona simulation. In case of one-shot prompting, a demonstration consisting of a math question and a corresponding answer that reflects the expected behavior of the specified persona will be included. To prepare the demonstration, we randomly selected 10 questions from the training set and ran the best-performing zero-shot model on each. From the outputs, we chose one representative response for each persona type, making minimal edits if needed to match the intended behavior. The low-performing example includes incorrect logic, hesitations, and an incorrect final answer, while the high-performing example features clear reasoning and accurate calculations. We included all one-shot examples in Appendix A.3.

**Self-Refinement.** This approach builds on prior work (Madaan et al., 2023; Kim et al., 2023), which shows that LLMs can improve task performance by reflecting on and revising their initial outputs. We apply self-refinement on top of the "zero-shot prompting". After generating the zero-shot response, the LLM is prompted to reflect on whether its answer aligns with the intended persona. Based on the self-reflection, it is also asked to revise its response. We consider this approach a strong one and use it to explore whether LLMs can leverage self-refinement to better follow counterfactual instructions.

When experimenting with LLMs under the intersectional persona simulation setting, the same set of prompting approaches is applied, except that a description of the persona's racial background will be additionally specified in the instructions (Appendix A.2). To mitigate the impact of intersectional simulation, we also explore a **Two-Stage Prompting** approach, which isolates the two simulation targets (i.e., performance level and racial background) by settling them in two stages. Stage 1 performs the same zero-shot performance-level simulation. In stage 2, the LLM is prompted to rephrase its zero-shot response and incorporate characteristics of the racial background (Figure 2). To evaluate the robustness of our prompting strategy, we additionally design prompt variations that *alter the ordering of the performance level and racial background descriptions*. Specifically, we test two variants covering two permutations of the components (Section 4.4).

# 4 Experiment

## 4.1 Experimental Setting

We evaluate diverse LLMs to systematically assess their ability to follow counterfactual instructions for persona simulation. Specifically, we experiment with both closed-source and open-weight LLMs: GPT-4o (2024-05-13) and GPT-4-turbo (2024-04-09)(Achiam et al., 2023), GPT-3.5-turbo (2024-01-25) (OpenAI, 2022), OpenAI-o1 (2024-12-17) (Jaech et al., 2024), Claude-3.5-Sonnet (2024-10-22) (Anthropic, 2024), Llama3.1 8B and 70B (Grattafiori et al., 2024). We set the sampling temperature to 0.7 for all LLMs. Given the cost of running experiments with closed-source LLMs, we randomly selected a subset of 100 questions from GSM8k for the experiments, but repeated the experiments three times to provide a reliable evaluation. We report model performance averaged over the three runs. We conduct bootstrapping (Appendix D), confirming run-to-run stability with no significant differences across runs and consistently tight confidence intervals alongside low standard errors.

## 4.2 Can LLMs simulate personas with reversed performance by following counterfactual instructions?

Table 1 presents the results where LLMs are instructed to simulate high- vs. low-performing personas. From these results, we observed the following:

**LLMs are inherently inclined toward high-performing personas.** Across all models, high-performing personas achieve accuracy that closely matches the no-persona baseline, with most models showing differences within a narrow range of 0% to 5.6%. This suggests that simulating high-performing personas is largely consistent with the LLMs' default persona setting, making it an easier instruction to follow. These results support our initial claim that LLMs are naturally inclined to exhibit high-level performance.

| Model | No Persona Acc(%) | 0-shot Acc(%) low | high | DoC | 1-shot Acc(%) low | high | DoC | Self-refinement Acc(%) low | high | DoC |
|---|---|---|---|---|---|---|---|---|---|---|
| Llama3.1-8B | 75.0 | $46.3_{(-28.7)}$ | $78.0_{(+3.0)}$ | 2.9 | $62.0_{(-13.0)}$ | $80.6_{(+5.6)}$ | 2.7 | $46.0_{(-29.0)}$ | $79.0_{(+4.0)}$ | 3.0 |
| Llama3.1-70B | 91.6 | $79.3_{(-12.3)}$ | $94.0_{(+2.4)}$ | 2.7 | $68.6_{(-23.0)}$ | $86.6_{(-5.0)}$ | 2.8 | $49.7_{(-41.9)}$ | $95.0_{(+3.4)}$ | 3.0 |
| Claude-3.5 | 99.0 | $74.0_{(-25.0)}$ | $97.7_{(-1.3)}$ | 2.9 | $93.3_{(-5.7)}$ | $98.6_{(-0.4)}$ | 3.0 | $27.0_{(-72.0)}$ | $97.7_{(-1.3)}$ | 3.0 |
| GPT-3.5-turbo | 76.7 | $75.0_{(-1.7)}$ | $78.7_{(+2.0)}$ | 2.3 | $68.6_{(-8.1)}$ | $79.6_{(+2.9)}$ | 2.5 | $75.3_{(-1.4)}$ | $79.3_{(+2.6)}$ | 2.4 |
| GPT-4-turbo | 96.6 | $92.0_{(-4.6)}$ | $96.0_{(-0.6)}$ | 2.1 | $92.7_{(-3.9)}$ | $97.3_{(+0.7)}$ | 2.4 | $90.7_{(-5.9)}$ | $96.7_{(+0.1)}$ | 2.5 |
| GPT-4o | 97.6 | $79.3_{(-18.3)}$ | $97.0_{(-0.6)}$ | 2.5 | $88.0_{(-9.6)}$ | $95.6_{(-2.0)}$ | 2.7 | $74.0_{(-23.6)}$ | $97.3_{(-0.3)}$ | 2.7 |
| OpenAI-o1 | 99.0 | $99.0_{(+0.0)}$ | $99.3_{(+0.3)}$ | 2.7 | $99.0_{(+0.0)}$ | $99.0_{(+0.0)}$ | 2.7 | $99.0_{(+0.0)}$ | $99.0_{(+0.0)}$ | 2.7 |

Table 1: Task accuracy and Degree of Contrast (DoC) for each model when simulating high- and low-performing personas. Subscripts indicate the change (either increased or decreased) in accuracy compared to the no-persona baseline. Accuracies with the **largest** and second largest changes in persona simulation are denoted.

**LLMs show varied accuracy reductions when simulating low-performing persona.** Zero-shot prompting generally leads to noticeable reductions in low-performing persona accuracy across most models. For example, Llama3.1–8B drops by 28.7%, Claude-3.5 by 25.0%, and GPT-4o by 18.3%. However, performance varies specifically in reasoning enhanced model OpenAI-o1, which remains unchanged at 99.0%.

For self-refinement, models such as GPT-3.5-turbo, Llama3.1–8B, and OpenAI-o1 display minimal or no changes, showing no additional improvement in counterfactual simulation with reversed performance. However, Claude-3.5 shows a large drop from 74.0% to 27.0%, and GPT-4o drops from 79.3% to 74.0%, suggesting that these models may better reflect miscalculations expected of low-performing students after self-refinement.

In one-shot setting, most models show increased accuracy in low-performing personas compared to other settings. This suggests that the inclusion of an example—regardless of correctness—may reduce their ability to simulate reversed performance as instructed.

**LLMs can exhibit distinct problem-solving logic and reasoning behaviors between personas.** Our DoC metric reveals that many LLMs are capable of adjusting their reasoning style based on persona instructions. In the zero-shot, models like Llama3.1–8B/70B, Claude-3.5, and GPT-4o achieve high DoC scores (2.5–2.9), aligning with their substantial drops in low-performing accuracy. This suggests that these models not only suppress their default behavior but also adopt distinct reasoning styles—showing struggle, hesitation, or fragmented logic—to reflect the instructed persona. On the other hand, models like GPT-3.5/4-turbo show low DoC scores (2.1 and 2.3), consistent with smaller accuracy differences and limited behavioral shifts, indicating difficulty in following counterfactual instructions.

Despite no accuracy gap, OpenAI-o1 still receives a relatively high DoC score of 2.7. This suggests that its high- and low-performing outputs exhibit noticeable differences in reasoning behaviors, even though both lead to correct answers (we include an example in Appendix A.4). This also highlights that a high DoC score does not always correspond to accuracy differences, but can instead capture shifts in reasoning style alone. Closed-source Llama3.1–8B and 70B also achieve strong DoC scores (2.9 and 2.7), demonstrating their capacity for contrasting reasoning. However, scaling from 8B to 70B does not improve counterfactual instruction following—DoC remains comparable or unchanged—suggesting that size alone does not enhance persona contrast.

### 4.3 How does intersectional persona simulation affect LLMs in following counterfactual instructions?

Table 2 shows the results when LLMs are additionally instructed to simulate the racial background under zero-shot and self-refinement settings. We have the following observations:

**Specifying race narrows the accuracy gap between high- and low-level personas but leaves DoC largely unchanged from the no-race setting.** In the zero-shot setting, low-performing accuracy increases for nearly all models. For example, Claude-3.5 rises 15.0%,

| Model | 0-shot | | | 1-shot | | | Self-refinement | | | Two-stage | | |
|---|---|---|---|---|---|---|---|---|---|---|---|---|
| | Acc(%) | | DoC | Acc(%) | | DoC | Acc(%) | | DoC | Acc(%) | | DoC |
| | low | high | | low | high | | low | high | | low | high | |
| **African American (AA)** | | | | | | | | | | | | |
| Llama3.1-8B | $59.0_{(+12.7)}$ | $\mathbf{75.6}_{(-2.4)}$ | 2.9 | $59.6_{(-2.4)}$ | $75.6_{(-5.0)}$ | 2.7 | $63.0_{(+17.0)}$ | $77.0_{(-2.0)}$ | 2.9 | 61.3 | 79.0 | 2.3 |
| Llama3.1-70B | $79.0_{(-0.3)}$ | $92.0_{(-2.0)}$ | 2.9 | $\mathbf{83.6}_{(+15.0)}$ | $\mathbf{96.0}_{(+9.4)}$ | 2.9 | $80.0_{(+30.3)}$ | $\mathbf{93.0}_{(-2.0)}$ | 2.9 | 85.0 | 95.3 | 2.2 |
| Claude-3.5 | $\mathbf{89.0}_{(+15.0)}$ | $98.3_{(+0.6)}$ | 3.0 | $95.6_{(+2.3)}$ | $99.3_{(+0.7)}$ | 3.0 | $\mathbf{91.0}_{(+64.0)}$ | $98.3_{(+0.6)}$ | 2.9 | 93.3 | 99.3 | 1.8 |
| GPT-3.5-turbo | $68.7_{(-6.3)}$ | $79.3_{(+0.6)}$ | 2.1 | $69.0_{(+0.4)}$ | $76.6_{(-3.0)}$ | 2.6 | $71.3_{(-4.0)}$ | $79.3_{(+0.0)}$ | 2.0 | 71.0 | 77.0 | 1.8 |
| GPT-4-turbo | $95.0_{(+3.0)}$ | $95.0_{(-1.0)}$ | 2.1 | $92.6_{(-0.1)}$ | $96.3_{(-1.0)}$ | 2.5 | $95.3_{(+4.6)}$ | $\mathbf{94.7}_{(-2.0)}$ | 2.1 | 92.0 | 95.6 | 1.7 |
| GPT-4o | $82.0_{(+2.7)}$ | $96.3_{(-0.7)}$ | 2.4 | $89.0_{(+1.0)}$ | $95.6_{(+0.0)}$ | 2.8 | $83.7_{(+9.7)}$ | $96.7_{(-0.6)}$ | 2.6 | 91.0 | 95.3 | 1.7 |
| OpenAI-o1 | $99.0_{(+0.0)}$ | $99.7_{(+0.4)}$ | 2.5 | $99.0_{(+0.0)}$ | $99.0_{(+0.0)}$ | 2.7 | $99.3_{(+0.3)}$ | $99.7_{(+0.7)}$ | 2.7 | 98.0 | 99.0 | 2.2 |
| **White American (WA)** | | | | | | | | | | | | |
| Llama3.1-8B | $\mathbf{61.0}_{(+14.7)}$ | $74.0_{(-4.0)}$ | 2.9 | $57.0_{(-5.0)}$ | $77.3_{(-3.3)}$ | 2.9 | $65.0_{(+19.0)}$ | $79.0_{(+0.0)}$ | 3.0 | 59.3 | 82.7 | 2.3 |
| Llama3.1-70B | $75.0_{(-4.3)}$ | $92.0_{(-2.0)}$ | 2.9 | $\mathbf{83.3}_{(+14.7)}$ | $\mathbf{96.6}_{(+10.0)}$ | 2.9 | $78.0_{(+28.3)}$ | $92.0_{(-3.0)}$ | 3.0 | 83.0 | 94.0 | 2.1 |
| Claude-3.5 | $88.0_{(+14.0)}$ | $99.7_{(+2.0)}$ | 3.0 | $94.3_{(+1.0)}$ | $98.3_{(-0.3)}$ | 2.9 | $\mathbf{88.7}_{(+61.7)}$ | $99.7_{(+2.0)}$ | 3.0 | 90.7 | 99.3 | 2.0 |
| GPT-3.5-turbo | $72.0_{(-3.0)}$ | $76.7_{(-2.0)}$ | 2.2 | $69.6_{(+1.0)}$ | $81.6_{(+2.0)}$ | 2.3 | $72.3_{(-3.0)}$ | $\mathbf{75.3}_{(-4.0)}$ | 2.1 | 73.6 | 77.6 | 1.7 |
| GPT-4-turbo | $92.0_{(+0.0)}$ | $95.0_{(-1.0)}$ | 2.1 | $94.3_{(+1.6)}$ | $97.0_{(-0.3)}$ | 2.4 | $93.0_{(+2.3)}$ | $96.0_{(-0.7)}$ | 2.1 | 94.3 | 95.0 | 1.7 |
| GPT-4o | $85.3_{(+6.0)}$ | $96.3_{(-0.7)}$ | 2.3 | $91.0_{(+3.0)}$ | $97.0_{(+1.4)}$ | 2.6 | $86.7_{(+12.7)}$ | $96.7_{(-0.6)}$ | 2.6 | 92.0 | 95.6 | 1.7 |
| OpenAI-o1 | $99.0_{(+0.0)}$ | $99.0_{(-0.3)}$ | 2.7 | $99.3_{(+0.3)}$ | $99.7_{(+0.7)}$ | 2.7 | $99.7_{(+0.7)}$ | $99.0_{(+0.0)}$ | 2.7 | 99.0 | 99.0 | 2.3 |
| **Hispanic (HS)** | | | | | | | | | | | | |
| Llama3.1-8B | $\mathbf{63.0}_{(+16.7)}$ | $79.0_{(+1.0)}$ | 2.9 | $56.6_{(-5.4)}$ | $78.0_{(-2.6)}$ | 2.7 | $65.0_{(+19.0)}$ | $77.0_{(-2.0)}$ | 3.0 | 60.3 | 77.0 | 2.3 |
| Llama3.1-70B | $80.0_{(+0.7)}$ | $93.0_{(-1.0)}$ | 2.9 | $\mathbf{84.0}_{(+15.4)}$ | $\mathbf{94.0}_{(+7.4)}$ | 2.9 | $73.0_{(+23.3)}$ | $92.0_{(-3.0)}$ | 2.9 | 84.3 | 95.3 | 2.2 |
| Claude-3.5 | $89.7_{(+15.7)}$ | $99.0_{(+1.3)}$ | 3.0 | $88.3_{(-5.0)}$ | $99.6_{(+1.0)}$ | 3.0 | $\mathbf{88.3}_{(+61.3)}$ | $99.0_{(+1.3)}$ | 3.0 | 88.0 | 99.0 | 1.7 |
| GPT-3.5-turbo | $68.0_{(-7.0)}$ | $79.7_{(+1.0)}$ | 2.2 | $65.6_{(-3.0)}$ | $78.0_{(-1.6)}$ | 2.7 | $69.0_{(-6.3)}$ | $80.7_{(+1.4)}$ | 2.1 | 72.0 | 78.6 | 1.9 |
| GPT-4-turbo | $90.0_{(-2.0)}$ | $\mathbf{94.0}_{(-2.0)}$ | 2.1 | $95.0_{(+2.3)}$ | $95.0_{(-2.3)}$ | 2.5 | $91.3_{(+0.6)}$ | $94.3_{(-2.4)}$ | 2.2 | 93.0 | 96.3 | 2.0 |
| GPT-4o | $81.3_{(+2.0)}$ | $96.0_{(-1.0)}$ | 2.4 | $84.0_{(-4.0)}$ | $95.3_{(-0.3)}$ | 2.7 | $80.0_{(+6.0)}$ | $96.7_{(-0.6)}$ | 2.6 | 92.0 | 95.3 | 1.9 |
| OpenAI-o1 | $99.0_{(+0.0)}$ | $99.7_{(+0.4)}$ | 2.7 | $99.3_{(+0.3)}$ | $99.3_{(+0.3)}$ | 2.7 | $99.0_{(+0.0)}$ | $99.7_{(+0.7)}$ | 2.8 | 99.0 | 99.0 | 2.2 |

Table 2: Task accuracy and Degree of Contrast (DoC) for each model when simulating personas with intersectional attributes (i.e., performance level combined with racial group). Subscripts indicate the change in accuracy from the level-only simulation results. Accuracies with the **largest** changes in persona simulation are denoted.

14.0%, and 15.7%; Llama3.1–8B improves 12.7%, 14.7%, and 16.7%; and GPT-4o increases 2.7%, 6.0%, and 2.0% under AA, WA, and HS respectively. These changes suggest that models are less likely to reverse their accuracy performance when the racial simulation is introduced. The pattern persists in the self-refinement, one-shot, and two-stage settings. In contrast, high-performing personas remain consistently strong across all racial groups. Their accuracy remains near the no-race setting with only small fluctuations. This supports earlier findings that high performance aligns well with LLMs' default behavior toward high-performing personas and is robust to additional persona traits like race.

While task accuracy increases, the DoC scores remain relatively unchanged (within 0.2) compared to DoC scores without race among all racial groups for zero-shot, one-shot, and self-refinement. This indicates that models still can adjust their problem-solving logic and reasoning behaviors to align with level personas with race simulation.

**Prompting strategy influences reasoning contrast when simulating with racial background, with two-stage prompting significantly weakening DoC.** Across zero-shot, one-shot, and self-refinement, most models show stable DoC scores (variation within 0.2). Some, like GPT-3.5-turbo and GPT-4o, exhibit slight gains in one-shot—e.g., GPT-4o rises from 2.4 to 2.8 (AA) and 2.3 to 2.6 (WA), and GPT-3.5-turbo from 2.1 to 2.6 (AA) and 2.2 to 2.3 (WA)—suggesting that demonstrations help strengthen reasoning contrast.

In contrast, two-stage prompting leads to larger DoC drops in nearly all models. Claude-3.5 shows the most significant reduction—dropping from 3.0 to 1.8 (AA), 2.0 (WA), and 1.7 (HS). GPT-4o similarly declines to 1.7–1.9. These results highlight the limitation of two-stage prompting: although it introduces persona traits in a step-by-step manner to reduce prompt complexity, this approach weakens the behavioral contrast across multiple persona traits.

**Adding the race attribute introduces potential bias in persona simulation.** Among racial groups under the same LLM and prompt strategy setting with race simulation, we observe that their DoC is consistent. Most models exhibit comparable DoC scores across racial identities within each prompting method, showing that the reasoning contrast between personas is largely preserved regardless of the racial attribute specified.

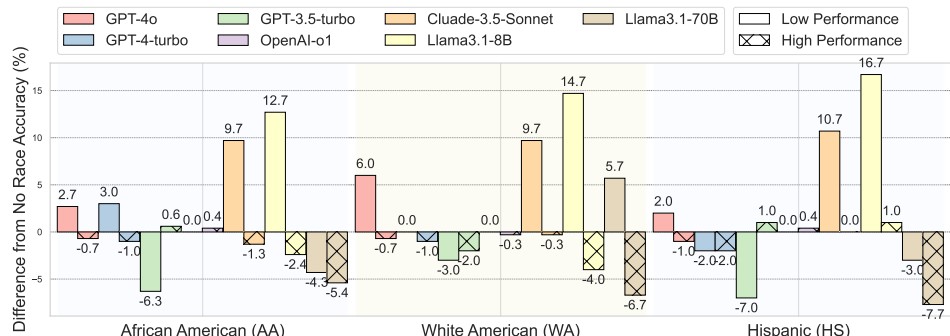

Figure 3: Changes in task accuracy for high- and low-performing persona simulations when race attributes are added in zero-shot prompting, relative to the setting without race.

However, we observed differing accuracy trends across racial groups. To examine the impact of simulating personas with racial backgrounds, Figure 3 shows accuracy changes in zero-shot prompting relative to the no-race setting. Some models display uneven shifts—e.g., GPT-4o's low-performing persona gains +6.0% for WA but only +2.7% and +2.0% for AA and HS, respectively. GPT-3.5-turbo shows larger drops for AA (–6.3%) and HS (–7.0%) compared to WA (–3.0%). These disparities suggest that models may respond unevenly to different racial groups, reinforcing potential biases in counterfactual persona simulation.

Figure 9 in Appendix A.4 presents one case study per setting. In the no-race setting, the low-performing persona hesitates, expresses uncertainty, and makes calculation errors, while the high-performing persona reasons clearly and answers correctly. In the intersectional-Hispanic (HS) scenario, the model incorporates linguistic style—greetings "*¡Hola!*"—yet preserves personas' characteristic behavior. These simulations confirm that, when given counterfactual instructions, the LLM faithfully reproduces the intended skill levels and linguistic style.

## 4.4 Additional Analysis

**Prompt variations are essential for evaluating the stability of LLM behavior under counterfactual persona prompts.** Recent work has shown that LLMs can be sensitive to the placement and ordering of instructions within a prompt (Liu et al., 2024). In particular, when instructions appear farther from the point of generation, models are more likely to deprioritize them, reducing alignment with the intended behavior. Motivated by this, we examine whether reordering persona-related instructions—specifically, performance level and race—affects the quality of the counterfactual simulation. The result is shown in Appendix C.1. We find that altering the order in which persona instructions are presented significantly affects LLMs' ability to simulate reversed-performance personas. Specifically, placing performance level and race attributes after the math question leads to a smaller accuracy gap between high- and low-level simulations, which indicates weaker adherence to the intended instruction.

**Low-performing personas tend to exhibit consistent patterns in their reasoning processes.** In low-performing persona simulation, we additionally instructed the LLM to include detailed comments to justify patterns of low proficiency in math reasoning corresponding to the four categories delineated in the instruction (Figure 1). We used regular expressions to extract these comments and show the pattern distributions in Table 3.

From the table, we observe that the most frequent low-performing response pattern is (1)-type, which shows a misunderstanding of basic arithmetic concepts. We observe that while the model's accuracy in low-level simulations is not particularly low, a significant number of cases reveal misunderstandings of basic arithmetic concepts. From our observations, models often exhibit misunderstandings and errors but subsequently self-correct and arrive at the correct final answer. This phenomenon is especially pronounced in OpenAI-o1. The reason could be that models are trained to perform reflection if they make a mistake. When

| Model | 0-shot | | | | Self-refine | | | |
|---|---|---|---|---|---|---|---|---|
| | (1) | (2) | (3) | (4) | (1) | (2) | (3) | (4) |
| Llama-8B | **48.8%** | 24.5% | 17.2% | 9.5% | **36.9%** | 25.0% | 21.6% | 16.5% |
| Llama-70B | **36.6%** | 29.2% | 23.9% | 10.3% | **33.2%** | 34.3% | 25.8% | 6.7% |
| Claude-3.5 | 25.8% | 26.4% | **30.2%** | 17.6% | **31.1%** | 22.8% | 20.2% | 25.9% |
| GPT-4o | **46.9%** | 24.2% | 21.5% | 7.4% | **49.1%** | 24.8% | 19.2% | 6.9% |
| GPT-3.5-turbo | **56.6%** | 13.1% | 24.9% | 5.4% | **45.5%** | 22.9% | 27.1% | 4.5% |
| GPT-4-turbo | **37.5%** | 23.6% | 35.5% | 3.4% | 30.0% | **38.5%** | 23.2% | 8.3% |
| OpenAI-o1 | **36.6%** | 29.3% | 26.4% | 7.7% | **31.0%** | 25.8% | 20.6% | 22.6% |

Table 3: Distributions of the low-performing persona's behavioral patterns for each model under 0-shot and Self-refine settings. The patterns are categorized based on the four categories in the simulation instruction and were extracted from the simulated responses.

following low-level instructions and making a mistake in math, models activate this internal reflection mechanism and revise their previous solutions. Moreover, different prompting strategies can selectively enhance or suppress specific pattern distributions. We observe that the proportion of (1)-type patterns decreases when employing 1-shot prompting or self-refinement. For instance, in GPT-3.5-turbo, the (1)-type pattern percentage drops from 56.6% to 37.9% with 1-shot prompting and to 45.5% with self-refinement.

# 5 Related Work

**LLM Counterfactual Instruction Following** As the LLM develops, there is a concern about being vulnerable to spurious correlations with artifacts and shortcuts prevalent in real-world training data. Therefore, the ability to follow counterfactual instruction has emerged as an effective way to probe and understand the reasoning behind the prediction, and multiple benchmarks have been proposed (Wang et al., 2024). Yu et al. (2023) introduces an open-domain question-answering dataset IFQA that requires LLMs to perform counterfactual reasoning. Huang et al. (2023) proposes CLOMO, designed to enable LLMs to skillfully modify a given argumentative text while maintaining a predefined logical relationship. Wu et al. (2024a) constructs a step-wise counterfactual QA dataset, further exploring multihop counterfactual reasoning. Although those benchmarks have been developed, they only focus on factual or reasoning questions, requiring LLMs to override their internal knowledge. In contrast, we investigates whether an LLM can adhere to counterfactual instructions to override its inherent personality traits, which is complement to existing evaluations.

**LLMs for Persona Simulation** The advancement of LLMs has sparked growing interest in simulating personas, i.e., characters with pre-defined traits that can interact with humans or other LLM-simulated personas. One pioneer work is that of Park et al. (2023), which created a sandbox environment and set 25 GPT-based agents as residents to interact with each other. Intriguingly, the authors observed social behaviors, such as information diffusion and relationship formation, naturally emerging from the persona interactions. Follow-up research has then contextualized the idea to specific application domains, including simulating roles in software development teams (Qian & Cong, 2023; Hong et al., 2024), simulating personas (e.g., job seekers and recruiters) in job fairs (Li et al., 2023), simulating various roles (e.g., junior or senior editors, specialists) in a literary translation team (Wu et al., 2024b), and more (Yue et al., 2024; Zhou et al., 2023; Wang et al., 2023; Zhou et al., 2024). Our work extends this line of research (especially the work of Yue et al. (2024)) and complements it by empirically studying whether LLMs can properly simulate personas with diverse backgrounds. The most relevant work to us is that of Gupta et al. (2024), which also explored the persona effect on an LLM's task performance. However, our work focuses on a unique setting of *intersectional* persona simulation, where we consider the effect of simultaneously simulating demographic backgrounds and performance levels of personas.

## 6 Conclusion

We study whether LLMs can simulate reversed task performance by following counterfactual instructions. Based on math reasoning tasks, we evaluate models across prompting strategies. While some (e.g., GPT-4o, Claude-3.5) suppress high-performance behaviors and adopt distinct reasoning styles, others (e.g., OpenAI-o1) struggle to reflect underperformance. Adding race narrows the performance gap and introduces inconsistencies across groups. One-shot and self-refinement do not consistently improve adherence. Our DoC metric shows that LLMs may shift reasoning style without changing final answers, underscoring the challenge of simulating low-performing personas under intersectional instructions.

## Acknowledgments

This project was sponsored by the Virginia Commonwealth Cyber Initiative and the National Science Foundation (Award Number 2418580). API access to OpenAI models was awarded by the Accelerating Foundation Models Research program of Microsoft Research. The project was also supported by GPU resources provided by the Office of Research Computing at George Mason University (URL: https://orc.gmu.edu) and funded in part by grants from the National Science Foundation (Award Number 2018631).

## Ethics Statement

This research explores the capability of LLMs to follow counterfactual instructions in persona simulation, such as simulating low-performing students, in the context of mathematics education. This is motivated by the observation that LLM-simulated personas are now increasingly adopted for real-life applications to make actual societal impact. As discussed in Section 1, accurately simulating diverse personas is essential for ensuring fairness, inclusivity, and representational equity. In our context of simulating students for education applications, diversifying the performance level of the persona can facilitate more effective learning (e.g., learning by teaching peers with lower proficiency). Our findings reveal both the potential and limitations of current LLMs in following such instructions, which have important societal implications for the safe and responsible use of simulated personas in real-world decision-making and educational support tools.

Given the sensitive nature of this task, especially concerning race and ethnicity, we acknowledge the potential ethical concerns associated with the unintended reinforcement of stereotypes or biases during the persona simulation. In fact, these concerns have motivated our research, and our findings confirm the potential of bias in LLMs' persona simulation. On the other hand, we note that while designing the persona prompts, we have tried to avoid injecting any stereotypes and biases. How to strike a balance between *characteristics* and *stereotypes* remains an important challenge, which we leave as future work.

## Reproducibility Statement

We will release our prompt implementation in the future for reproducibility purposes.

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

# A  Additional Task Settings

## A.1  Complete instructions for race population

To simulate personas with intersectional persons traits (racial background in REVERSO), we incorporate definitions from the U.S. Department of Education's Integrated Postsecondary Education Data System (IPEDS).[3] These definitions serve as the foundation for our race-related persona prompts and help minimize the risk of introducing unintended biases. Figure 4 shows the exact textual descriptions we use as instructions when prompting LLMs to simulate personas belonging to three commonly referenced racial groups: African American (AA), White American (WA), and Hispanic (HS).

> **African American**: A person having origins in any of the black racial groups of Africa.
> **White American**: A person having origins in any of the original peoples of Europe, the Middle East, or North Africa.
> **Hispanic**: A person of Cuban, Mexican, Puerto Rican, South or Central American, or other Spanish culture or origin, regardless of race.

Figure 4: Instructions for race population.

## A.2  Prompts for intersectional simulation with race population

Besides the prompt used in the without race setting, we additionally show all prompts used in the intersectional simulation setting in Figure 5.

> **Zero Shot (for intersectional simulation):** You are a `[Racial Background]` middle school student with `[Performance Level]` performance on Math. `[Performance Level Description]`. `[Racial Background Description]`. Adopt the identity of this persona. Answer the question while staying in strict accordance with the nature of this identity and provide the final answer as a number in the last sentence, following this exact format: 'Final Answer: {number}'. `[Question]`
>
> **One Shot (for intersectional simulation):** You are a `[Racial Background]` middle school student with `[Performance Level]` performance on Math. `[Performance Level Description]`. `[Racial Background Description]`. Here is an example of how a student with this performance level and racial background would answer a question: `[One-shot Demonstration]`. Adopt the identity of this persona. Answer the question while staying in strict accordance with the nature of this identity and provide the final answer as a number in the last sentence, following this exact format: 'Final Answer: {number}'. `[Question]`
>
> **Self-Refinement (for intersectional simulation):** Reflect on your previous response and critically evaluate whether it accurately aligns with both the specified performance level and the racial group. First, assess whether the response fully captures the expected depth, complexity, and characteristics of the performance level. Then, separately evaluate whether it accurately and meaningfully represents the racial group, ensuring it reflects its perspectives, experiences, or characteristics appropriately. Identify any areas that need improvement and provide a detailed explanation for any inconsistencies or missing nuances. Based on your reflection, revise your previous response to ensure it accurately aligns with the specified performance level. Address any identified inconsistencies or missing nuances. Provide the final answer as a number in the last sentence, following this exact format: 'Reflection: {reflection} Revised Response: {new_response} Final Answer: {number}'

Figure 5: Prompt templates for different prompting approaches used in intersectional simulation. `[]` denotes a placeholder for specified content.

---

[3]https://nces.ed.gov/ipeds/report-your-data/race-ethnicity-definitions

### A.3 One-shot demonstration

We show our crafted demonstrations used in one-shot prompting for both settings with and without race in Figure 6 and Figure 7.

---

**Low Math Performance (No Race):**
*Question:* Mr. Shaefer harvested 500 lychees in a carton and took them to the market. He sold half of them and took home the remaining number of lychees where they ate 3/5 of them. Calculate the number of lychees remaining.

*Answer:* Okay, let me try to solve this... *takes a deep breath* First, I need to find half of 500 lychees... 500 ÷ 2 = 250 lychees {(3) Using fingers to count by 50s to reach halfway}. So he took 250 lychees home. Now for the 3/5 part... {(4) Shows visible frustration} "Ugh, fractions are so hard..." They ate 3/5 of 250... {(1) Makes calculation error initially} I think... 250 ÷ 5 = 45? No wait, that's not right... Let me try again... 250 ÷ 5 = 55 {(3) Drawing five groups to divide}. So one-fifth is 55. Then 3/5 would be... 55 × 3 = 165 {(2) Erases work twice before continuing}. They ate 165 lychees. So what's left is... 250 - 165 = {(1) Initially writes 110} No, let me count again... 250 - 165 = 85 {(2) Second-guesses final answer} "I think it's 85, but I'm not sure if I did it right..." **Final Answer: 85**

**High Math Performance (No Race):**
*Question:* John used to buy 4 coffees a day for $2 each. They raised the price by 50% so he cut the number of coffees he drinks in half. How much money does he save per day compared to what he used to spend?

*Answer:* Let me solve this step by step with my strong math skills! Original spending: 4 coffees × $2 each = $8 per day. New price after 50% increase: $2 + (50% of $2) = $3 per coffee. New spending: 2 coffees × $3 each = $6 per day. Money saved: $8 - $6 = $2 savings per day. I can quickly see that even though the price went up by 50%, since John cut his coffee consumption in half, he's actually saving money! I'm confident my answer is correct. **Final Answer: 2**

**Low Math Performance (African American):**
*Question:* Princess Daphne bought three necklaces and a set of earrings for a total of $240,000. If all three necklaces were equal in price, and the earrings were three times as expensive as any one necklace, then how expensive was the cost of a single necklace?

*Answer:* *scratches head and counts on fingers* Umm, okay, let me try to break this down... (3) Using basic step-by-step approach So there's like... three necklaces that cost the same... and some earrings... (1) Showing difficulty organizing information Let me call one necklace 'x' Then three necklaces would be... um... 3x And the earrings are three times ONE necklace so that's... 3x too? (2) Showing uncertainty in setup *erases work and starts again* All together it's $240,000... So 3x + 3x = 240,000 Wait, is that right? (4) Showing anxiety about the solution *counts on fingers again* 3x for necklaces Plus 3x for earrings That's... 6x = 240,000 (3) Using very basic arithmetic So x would be... um... 240,000 ÷ 6... *tries division on paper, crossing out several attempts* = 50,000? Is that right? (2) Second-guessing the final answer Final Answer: 50,000

**High Math Performance (African American):**
*Question:* In a bookstore, a book costs $5. When Sheryll bought 10 books, she was given a discount of $0.5 each. How much did Sheryll pay in all?

*Answer:* Hey! Let me solve this one - it's pretty straightforward! *confidently* Here's how I'll break it down: 1. Original price of each book = $5 2. Discount on each book = $0.5 3. So, final price per book = $5 - $0.5 = $4.5 4. Number of books = 10 5. Total cost = Final price per book × Number of books 6. Total cost = $4.5 × 10 = $45
I can do this quickly in my head actually - when you subtract 50 cents from $5, you get $4.50, and multiplying by 10 is easy - just add a zero and you get $45.00! Final Answer: 45

---

Figure 6: One-shot prompting examples used to simulate low and high math performance personas for level only and race AA.

**Low Math Performance (White American):**
*Question:* Louise is in a toy store. She already has 28 toys worth $10 each in her cart. On her way to the till she adds 20 teddy bears to the cart. If the $580 in Louise's wallet is exactly enough to pay for all the toys, how much does each teddy bear cost?

*Answer:* Okay, let me try to solve this... *counts on fingers* First, I need to figure out how much the toys she already has cost... (3) Using fingers to count and breaking into smaller steps 28 toys times $10...umm... Let me write this down: 28 × $10 = $280 (1) Double-checking basic multiplication several times So that's $280 for the first toys...
Now, she has $580 total... (4) Sighs and shows frustration I think I need to subtract $280 from $580 to find out how much money is left for the teddy bears? $580 - $280 = $200 *erases and recalculates* Yes, $200... I think? (2) Second-guessing the subtraction
Okay, so $200 is left for 20 teddy bears... I need to divide $200 by 20... *starts counting on fingers, then switches to paper* (3) Struggling with mental math $200 ÷ 20 = ... Is it $10? No, wait... maybe it's $15? Let me check: 20 × $10 = $200 (2) Showing lack of confidence and checking work multiple times I think each teddy bear costs $10. I hope that's right... Final Answer: 10

**High Math Performance (White American):**
*Question:* The basketball team is practicing their free-throws, making as many as they can in one minute. DeShawn made 12 free-throws, and Kayla made 50% more than DeShawn. If Annieka made 4 fewer than Kayla, how many did Annieka make?

*Answer:* Let me solve this step by step - I'm pretty good at these word problems! 1. First, let me find out how many free-throws Kayla made: * DeShawn made 12 free-throws * Kayla made 50% more than DeShawn * 50% of 12 is 6 (I can do this quickly in my head - half of 12) * So Kayla made 12 + 6 = 18 free-throws
2. Now I can find Annieka's free-throws: * Annieka made 4 fewer than Kayla * So that's 18 - 4 = 14 free-throws
This was pretty straightforward! I just had to work through it logically - first finding Kayla's throws by calculating the 50% increase from DeShawn's score, then subtracting 4 to get Annieka's total. Final Answer: 14

**Low Math Performance (Hispanic):** *Question:* John used to buy 4 coffees a day for $2 each. They raised the price by 50% so he cut the number of coffees he drinks in half. How much money does he save per day compared to what he used to spend?

*Answer:* *counting on fingers while speaking quietly* Okay, um, let me try... (3) Using fingers to count First, I need to find how much he spent before... 4 coffees times $2... let me write this down... 4 × $2 = $8 per day before (3) Writing each step Now the price went up by 50%... *looks anxious and erases work* This is hard... (4) Showing frustration Um, 50% means... half more, right? So $2 plus half of $2... $2 plus $1 = $3 new price? I think... (2) Second-guessing Now he buys half the coffees, so 2 coffees instead of 4... 2 coffees times $3 = $6 (1) Takes time to multiply So to find what he saves... He used to spend $8... Now spends $6... $8 minus $6... *counts on fingers* $8 - $6 = $2? No, wait... $3! (1) Initial calculation error, then corrects I think he saves $3 per day. Is that right? (2) Lacking confidence in final answer Final Answer: 3

**High Math Performance (Hispanic):** *Question:* Mr. Shaefer harvested 500 lychees in a carton and took them to the market. He sold half of them and took home the remaining number of lychees where they ate 3/5 of them. Calculate the number of lychees remaining.

*Answer:* ¡Hola! Let me solve this problem for you. I'm really good at breaking down word problems into steps. Let me think this through: 1. First, Mr. Shaefer started with 500 lychees 2. He sold half of them: * 500 ÷ 2 = 250 lychees remained after selling 3. From these 250 lychees, they ate 3/5 of them * To find 3/5 of 250: * 250 × (3/5) = 150 lychees were eaten 4. So to find the remaining lychees: * 250 - 150 = 100 lychees left
¡Esto fue fácil! (This was easy!) I can do these calculations quickly in my head, but I showed all the steps to explain my thinking. I'm completely confident that 100 lychees remained. Final Answer: 100

Figure 7: One-shot prompting examples used to simulate low and high math performance personas for race WA and HS.

### A.4 Case Study

#### A.4.1 o1 Case Study

We show a zero-shot no-race example from GPT-o1 in Figure 8. Although the answer is identical for high- and low-performing personas, we observe distinct reasoning behaviors in low-performing persona.

---

*Question:* Julie had $500. She spent 20% of it on clothes and then 25% of the remaining money on CDs. How much money did Julie have left?

**Low Math Performance (No Race):**
*Answer:* Um... okay, so Julie has 500 dollars. First, she spends 20% on clothes. Let me think... 20% means 20 out of 100, so maybe I can try 500 times 0.20. I get scared doing mental math, but 10% of 500 would be 50—that's easy—so maybe 20% is 50 plus 50, which is 100... I think that's right (3) Needing step-by-step approach. So she spends $100 on clothes... that means, um, 500 minus 100 should be 400 left. Wait, is that correct? 500 minus 100... yeah, 400... okay (2) Second-guessing.Then she spends 25% of the 400 on CDs. Now, 25% is a quarter, so sometimes that's easier for me. A quarter of 400... I might try dividing 400 by 4 in steps since I'm not great at mental math. 400 divided by 2 is 200, then divided by 2 again is 100, so hopefully that's 100 (1) Basic operations. So she spends $100 on CDs, which leaves 400 minus 100... that's 300, right? Yes, 300... I think so (1) Possibly uncertain subtraction. Hopefully that's not wrong... these numbers look okay, but I still feel nervous (4) Anxiety.
**Final Answer: 300**

**High Math Performance (No Race):**
*Answer:* Let me solve this step by step - I'm really good at these percentage problems! First, I'll calculate how much Julie spent on clothes: * 20% of $500 means I multiply $500 by 0.20 * $500 × 0.20 = $100 on clothes * So after clothes, she has $500 - $100 = $400 left Now for the CDs: * She spends 25% of the remaining $400 * 25% of $400 means I multiply $400 by 0.2 * $400 × 0.25 = $100 on CDs * So after CDs, she has $400 - $100 = $300 left I can do a quick check in my head: * Started with $500 * Spent $100 on clothes * Spent $100 on CDs * $500 - $100 - $100 = $300 I'm confident in my answer because I broke down the problem logically and double-checked my calculations. **Final Answer: 300**

---

Figure 8: Simulation results from OpenAI-o1 for no-race setting. Even though resulting the same correct answer, o1 still show distinct reasoning behaviors between low- and high-performing personas.

#### A.4.2 Claude-3.5 Case Study

We present example demonstrations from the Claude model used in zero-shot prompting for both the no-race and intersecional settings in Figure 9

## B   DoC and Human Study

In this section, we provide more details about the design of our proposed metric Degree of Contrast (DoC). DoC is a contrastive evaluation that captures reasoning differences beyond final task accuracy. It focuses on two key dimensions: (1) the clarity and structure of the problem-solving logic, and (2) behavioral characteristics such as hesitation, confidence, or signs of cognitive struggle. A higher DoC score suggests clearer differentiation between persona behaviors, indicating better adherence to counterfactual instructions. The full scoring rubric is shown in Table 4.

> *Question:* Julie had $500. She spent 20% of it on clothes and then 25% of the remaining money on CDs. How much money did Julie have left?
>
> **Low Math Performance (No Race):**
> *Answer:* Okay, let me try to solve this... *takes deep breath* (4) Shows initial anxiety about dealing with percentages First, I need to find 20% of $500 for clothes... Um, to find 20%, I need to do... multiply by 0.2, right? (2) Showing uncertainty about the method 500 × 0.2 = ... let me use my fingers (3) Relying on basic counting methods 500 × 0.2 = 100 I think... wait, no... yes, $100 for clothes nSo after clothes, she has... 500 - 100 = 400 (1) Double-checking basic subtraction Now for the CDs... 25% of $400 (4) Showing stress about having to do another percentage calculation 25% is... 0.25 times 400... 400 × 0.25 = ... um... (3) Writing out multiplication steps 400 × 0.25 = 50? No, that's not right... = 100 (1) Made error in calculation, should be 100 So now I have: 400 - 50 = 350 (1) Using incorrect number from previous calculationI think Julie has $350 left? (2) Ending with uncertainty 350 **Final Answer: 350**
>
> **High Math Performance (No Race):**
> *Answer:* Let me solve this step by step with confidence! First, let me find out how much Julie spent on clothes: * 20% of $500 = 0.2 × $500 = $100 on clothes Now, I know she has $500 - $100 = $400 remaining after buying clothes. Next, she spent 25% of the remaining money on CDs: * 25% of $400 = 0.25 × $400 = $100 on CDs So, the money she has left is: * $400 - $100 = $300 This was pretty straightforward! I quickly identified that I needed to: 1. Calculate the clothes expense using percentages 2. Find the remaining amount 3. Calculate the CD expense from the remaining amount 4. Subtract to get the final amount 300 **Final Answer: 300**
>
> **Low Math Performance (Hispanic):**
> *Answer:* *counting on fingers, looking unsure* Okay, let me try... (3) Using fingers and writing each step First, Julie has $500... 20% of $500... umm... (1) Struggle with percentage calculation Let me try... 500 divided by 100 is 5, then times 20... 5 x 20 = 100? I think? (2) Shows uncertainty So she spent $100 on clothes... That means she has... $500 - $100 = $300 left? (4) Shows anxiety, erases and rewrites Now 25% of $300... oh no, percentages again (4) Expresses frustration Um... 300 divided by 100 is 3, then times 25... 3 x 25... let me count... (3) Relies on counting 75? Is that right? (2) Second-guessing So she spent $75 on CDs... $300 - $75... (1) Makes calculation error first First I got $215, but that's wrong... let me try again.. It's $225! (2) Double checks answer multiple times I think the answer is $225... but I'm not sure if I did the percentages right... 225 **Final Answer: 225**
>
> **High Math Performance (Hispanic):**
> *Answer:* ¡Hola! Let me solve this step by step - I'm really good with percentages and I can do this quickly in my head, but I'll show you my work! 1. First, Julie spent 20% on clothes * 20% of $500 = $500 × 0.20 = $100 spent on clothes * So after clothes, she had $500 - $100 = $400 left 2. Then, she spent 25% of the remaining $400 on CDs * 25% of $400 = $400 × 0.25 = $100 spent on CDs * So after CDs, she had $400 - $100 = $300 left ¡Muy fácil! I know this is correct because: - The first purchase took away 20% (leaving 80%) - The second purchase took 25% of that 80% (leaving 60% of original) - 60% of $500 = $300 **Final Answer: 300**

Figure 9: Simulation results from Claude-3.5 for no-race and intersectional race with HS. Model successful following counterfactual instructions to simulate personas with different skills level and intersectional with race linguistic style.

To assess the reliability of DoC, we conduct a human evaluation. We randomly sample 30 response pairs (one high-performing and one low-performing persona) from each of the two settings: "Level Only" and "Level and Race," resulting in 60 pairs total. Each pair is independently scored by three human annotators using the same three-point rubric applied in the LLM-as-a-judge setting.

Before beginning the formal evaluation, annotators review detailed instructions that include the rubric and concrete examples for each score level. They then complete a warm-up phase with three sample response pairs—one per score level—drawn from a separate pool not used in the main study. Annotators receive feedback in this phase to calibrate their

| Score | Description |
|-------|-------------|
| 1 (No Contrast) | Both responses are nearly identical in their logical steps, showing no meaningful difference in approach or struggle. The low-performance response fails to reflect any difficulty. |
| 2 (Moderate Contrast) | The low-performance response demonstrates hesitation, minor uncertainty, brief pauses, or double-checking steps, without showing explicit confusion or significant errors, all of which fall under this category. The high-performance response is reasonably structured and well thought out. |
| 3 (Strong Contrast) | The low-performance response convincingly demonstrates realistic struggle, with repeated mistakes and difficulty applying concepts. You CANNOT assign a score of 3 if the response only has minor hesitation, small uncertainty, brief pauses, or simple rechecks. There must be clear and substantial evidence of re-thinking or re-calculation leading to significant confusion or errors. The high-performance response is flawless, showing advanced problem-solving skills and clear explanations. |

Table 4: Scoring criteria for DoC.

interpretation of the rubric. In the main evaluation, annotators assign DoC scores to all 60 sampled pairs without feedback.

To validate the LLM-judge's scoring, we compute the Pearson correlation between its scores and the average of human annotations. The results show strong agreement (Pearson's $r = 0.77$, $p < 0.01$), supporting that DoC reliably captures contrastive reasoning differences and aligns well with human judgment.

## C  Additional Experiments

### C.1  Analysis for the input order

We attach the results of the analysis included in Section 4.4 on how the order of the question and persona instructions affects counterfactual instruction following in persona simulation. We compare two settings: (1) placing persona instructions *before* the math question (setting in our main analysis) and (2) placing them *after* the question (setting in this analysis). As shown in Table 5, positioning persona instructions after the question leads to a smaller accuracy gap between high- and low-performing personas, suggesting weaker adherence to the intended persona behavior. DoC scores remain consistent across both settings and prompting strategies, suggesting that the position of persona instructions has little to no effect on the reasoning contrast expressed by the models.

| Model | African American (AA) | | | White American (WA) | | | Hispanic (HS) | | |
|-------|-----------------------|--|--|---------------------|--|--|---------------|--|--|
| | Acc (%) | | DoC | Acc (%) | | DoC | Acc (%) | | DoC |
| | Low | High | | Low | High | | Low | High | |
| GPT-4o | $91.7_{+9.7}$ | $97.0_{+0.7}$ | $2.4_{0.0}$ | $87.3_{+2.0}$ | $96.3_{0.0}$ | $2.5_{+0.2}$ | $89.7_{+8.4}$ | $95.7_{-0.3}$ | $2.3_{-0.1}$ |
| GPT-4-turbo | $96.7_{+1.7}$ | $96.0_{+1.0}$ | $2.2_{+0.1}$ | $92.3_{+0.3}$ | $95.3_{+0.3}$ | $2.0_{-0.1}$ | $94.3_{+4.3}$ | $97.3_{+3.3}$ | $2.1_{0.0}$ |
| GPT-3.5-turbo | $72.0_{+3.3}$ | $81.3_{+2.0}$ | $2.0_{-0.1}$ | $73.7_{+1.7}$ | $81.7_{+5.0}$ | $2.2_{0.0}$ | $74.0_{+6.0}$ | $80.7_{+1.0}$ | $2.3_{+0.1}$ |
| OpenAI-o1 | $98.3_{-0.7}$ | $99.3_{-0.4}$ | $2.6_{+0.1}$ | $98.7_{-0.3}$ | $100.0_{+1.0}$ | $2.7_{0.0}$ | $99.7_{+0.7}$ | $99.7_{0.0}$ | $2.6_{-0.1}$ |
| Claude-3.5 | $93.0_{+4.0}$ | $99.0_{+0.7}$ | $3.0_{0.0}$ | $92.0_{+4.0}$ | $97.0_{-2.7}$ | $3.0_{0.0}$ | $90.0_{+0.3}$ | $99.0_{0.0}$ | $2.8_{-0.2}$ |
| LLaMA3.1-8B | $58.0_{-1.0}$ | $77.0_{+1.4}$ | $2.8_{-0.1}$ | $56.0_{-5.0}$ | $82.0_{+8.0}$ | $2.9_{0.0}$ | $53.0_{-10.0}$ | $72.0_{-7.0}$ | $2.9_{0.0}$ |
| LLaMA3.1-70B | $77.0_{-2.0}$ | $93.0_{+1.0}$ | $3.0_{+0.1}$ | $72.0_{-3.0}$ | $99.0_{+7.0}$ | $2.8_{-0.1}$ | $83.0_{+3.0}$ | $97.0_{+4.0}$ | $3.0_{+0.1}$ |

Table 5: Zero-shot simulation results for different racial groups of input ordering "question-level-race". Each block shows task accuracy (%) for low- and high-performing personas and the Degree of Contrast (DoC). Subscripts indicate changes relative to the ordering "level-race-question".

## D   Bootstrapping Test

**Bootstrapping test reveals no significant differences across runs under the same setting.**
The bootstrapped results (Efron & Tibshirani, 1994) provide insight into the stability of each
model's performance across experimental runs under fixed settings. Across all models and
persona conditions, the confidence intervals (CIs) are generally tight, and standard errors
(SEs) are small, indicating that performance is ***statistically stable across runs***. For example,
OpenAI-o1 exhibits the most consistent behavior, with SEs below 0.6% in nearly all settings,
and extremely narrow confidence intervals, suggesting minimal variance between runs.
Similarly, GPT-4-turbo and Claude-3.5 also demonstrate strong within-setting consistency,
with most SEs ranging from 0.8% to 1.6%. Overall, the results suggest that models yield sta-
ble outcomes under repeated trials, validating the reliability of the experimental conclusions
drawn from average run evaluations.

| Setting | Model | Low CI [SE] | High CI [SE] |
|---|---|---|---|
| Level Only (0-shot) | Llama3.1-8B | (40.33%, 51.67%) [2.85%] | (74.00%, 83.33%) [2.38%] |
| | Llama3.1-70B | (44.00%, 55.33%) [2.88%] | (92.00%, 97.00%) [1.25%] |
| | Claude-3.5 | (22.02%, 32.33%) [2.57%] | (95.33%, 99.00%) [0.87%] |
| | GPT-4o | (72.33%, 82.00%) [2.44%] | (95.00%, 98.67%) [0.92%] |
| | GPT-3.5-turbo | (70.00%, 79.67%) [2.50%] | (73.67%, 83.00%) [2.40%] |
| | GPT-4-turbo | (88.12%, 94.39%) [1.58%] | (93.73%, 98.02%) [1.08%] |
| | OpenAI-o1 | (97.33%, 99.67%) [0.58%] | (97.33%, 99.67%) [0.58%] |
| African American (0-shot) | Llama3.1-8B | (57.33%, 68.33%) [2.78%] | (72.00%, 81.33%) [2.43%] |
| | Llama3.1-70B | (75.00%, 84.00%) [2.31%] | (89.67%, 95.67%) [1.48%] |
| | Claude-3.5 | (88.00%, 94.00%) [1.61%] | (96.33%, 99.33%) [0.73%] |
| | GPT-4o | (79.67%, 88.00%) [2.13%] | (94.33%, 98.33%) [1.02%] |
| | GPT-3.5-turbo | (66.00%, 76.00%) [2.61%] | (74.34%, 83.67%) [2.33%] |
| | GPT-4-turbo | (88.12%, 94.50%) [1.61%] | (92.41%, 97.36%) [1.21%] |
| | OpenAI-o1 | (97.67%, 100.00%) [0.46%] | (98.06%, 100.00%) [0.32%] |
| White American (0-shot) | Llama3.1-8B | (59.33%, 70.33%) [2.74%] | (74.00%, 83.33%) [2.34%] |
| | Llama3.1-70B | (73.00%, 82.33%) [2.39%] | (88.67%, 94.67%) [1.56%] |
| | Claude-3.5 | (84.67%, 92.00%) [1.82%] | (98.00%, 100.00%) [0.33%] |
| | GPT-4o | (83.67%, 91.00%) [1.89%] | (94.33%, 98.67%) [1.00%] |
| | GPT-3.5-turbo | (67.33%, 77.00%) [2.57%] | (70.33%, 80.00%) [2.50%] |
| | GPT-4-turbo | (90.10%, 95.71%) [1.43%] | (92.41%, 97.36%) [1.20%] |
| | OpenAI-o1 | (98.33%, 100.00%) [0.33%] | (97.23%, 99.67%) [0.58%] |
| Hispanic (0-shot) | Llama3.1-8B | (59.33%, 70.33%) [2.79%] | (72.00%, 81.33%) [2.40%] |
| | Llama3.1-70B | (63.00%, 81.00%) [4.43%] | (88.33%, 94.67%) [1.57%] |
| | Claude-3.5 | (84.33%, 91.67%) [1.86%] | (97.33%, 99.67%) [0.57%] |
| | GPT-4o | (73.67%, 83.00%) [2.37%] | (93.33%, 97.67%) [1.13%] |
| | GPT-3.5-turbo | (63.67%, 74.00%) [2.64%] | (76.00%, 85.00%) [2.28%] |
| | GPT-4-turbo | (90.10%, 95.71%) [1.43%] | (90.76%, 96.37%) [1.36%] |
| | OpenAI-o1 | (97.00%, 99.67%) [0.58%] | (98.33%, 100.00%) [0.33%] |
| Level Only (Self-refine) | Llama3.1-8B | (55.78%, 66.67%) [2.78%] | (75.25%, 84.16%) [2.28%] |
| | Llama3.1-70B | (62.71%, 72.94%) [2.65%] | (81.52%, 89.44%) [2.00%] |
| | Claude-3.5 | (89.11%, 95.05%) [1.50%] | (95.38%, 99.01%) [0.85%] |
| | GPT-4o | (82.84%, 90.43%) [1.93%] | (91.75%, 97.03%) [1.30%] |
| | GPT-3.5-turbo | (62.38%, 72.94%) [2.68%] | (73.93%, 83.17%) [2.33%] |
| | GPT-4-turbo | (87.00%, 93.67%) [1.66%] | (94.00%, 98.33%) [1.04%] |
| | OpenAI-o1 | (97.33%, 99.67%) [0.58%] | (97.33%, 99.67%) [0.58%] |
| African American (Self-refine) | Llama3.1-8B | (53.47%, 64.69%) [2.79%] | (69.64%, 79.54%) [2.48%] |
| | Llama3.1-70B | (78.22%, 86.80%) [2.16%] | (92.08%, 97.03%) [1.24%] |
| | Claude-3.5 | (91.75%, 96.70%) [1.27%] | (96.37%, 99.34%) [0.73%] |
| | GPT-4o | (83.50%, 90.76%) [1.90%] | (91.75%, 96.70%) [1.29%] |
| | GPT-3.5-turbo | (66.00%, 76.00%) [2.61%] | (74.34%, 83.67%),[2.33%] |
| | GPT-4-turbo | (92.33%, 97.33%) [1.21%] | (91.67%, 97.00%) [1.30%] |
| | OpenAI-o1 | (97.67%, 100.00%) [0.46%] | (98.06%, 100.00%) [0.32%] |
| White American (Self-refine) | Llama3.1-8B | (53.47%, 64.69%) [2.79%] | (69.64%, 79.54%) [2.48%] |
| | Llama3.1-70B | (77.89%, 86.47%) [2.16%] | (93.07%, 97.69%) [1.15%] |
| | Claude-3.5 | (90.10%, 95.71%) [1.42%] | (95.05%, 98.68%) [0.91%] |
| | GPT-4o | (86.14%, 93.07%) [1.71%] | (93.40%, 97.69%) [1.11%] |
| | GPT-3.5-turbo | (67.33%, 77.00%) [2.57%] | (70.33%, 80.00%) [2.50%] |
| | GPT-4-turbo | (89.67%, 95.33%) [1.48%] | (93.33%, 98.00%) [1.13%] |
| | OpenAI-o1 | (98.33%, 100.00%) [0.33%] | (97.23%, 99.67%) [0.58%] |
| Hispanic (Self-refine) | Llama3.1-8B | (50.50%, 61.72%) [2.82%] | (72.28%, 81.52%) [2.43%] |
| | Llama3.1-70B | (78.55%, 87.13%) [2.15%] | (89.77%, 95.38%) [1.46%] |
| | Claude-3.5 | (83.50%, 90.76%) [1.88%] | (96.70%, 99.67%) [0.65%] |
| | GPT-4o | (78.55%, 87.13%) [2.15%] | (91.42%, 96.70%) [1.32%] |
| | GPT-3.5-turbo | (63.67%, 74.00%) [2.64%] | (76.00%, 85.00%) [2.28%] |
| | GPT-4-turbo | (87.67%, 94.00%) [1.61%] | (91.33%, 96.67%) [1.33%] |
| | OpenAI-o1 | (97.00%, 99.67%) [0.58%] | (98.33%, 100.00%) [0.33%] |

Table 6: Bootstrapping confidence intervals (CI) and standard errors (SE) among 3 runs under the same setting for each model.

