# OpenReview forum: "Can LLMs Simulate Personas with Reversed Performance? A Systematic Investigation for Counterfactual Instruction Following in Math Reasoning Context"
_colmweb.org/COLM/2025/Workshop/Social_Sim — Social Sim'25_

### Official Review · Reviewer_risx · 2025-07-16
**Can LLMs Simulate Personas with Reversed Performance? A Systematic Investigation for Counterfactual Instruction Following in Math Reasoning Context**

**Rating:** 8
**Overall Assessment:** 4
**Confidence:** 5

**Review:**

The paper addresses a novel and timely question: is it possible to induce an LLM to act like an underperforming persona through instructions? The authors formalize this notion as counterfactual instruction following, operationalized as performing math reasoning with different personas. A wide variety of prompting strategies and LLMs (closed and open source) are evaluated and a new metric (Degree of Contrast, DoC) for capturing the difference in reasoning beyond mere accuracy is proposed.
The paper is well written, with a sound methodology, and the experiments are designed and reported appropriately. The experiments look beyond accuracy and consider actual reasoning behavior. The usage of intersectional traits (race) is a nice touch, which naturally reveals existing gaps in fairness. Ethical considerations are well-articulated.
Two more additions could have strengthened the paper: the discussion of why some models (OpenAI o1, for example) do not show decreased accuracy even though different behavior is observed, and more potential real-world applications or downstream use cases.
Pros :
  * Proposes a new concept: counterfactual instruction following
  * Solid experimental design and broad model coverage
  * Introduces DoC to go beyond right/wrong answers
  * Considers intersectional attributes (e.g., race)
  * Includes ethical reflection and human validation

Cons :
  * Accuracy remains high in some cases even when simulating low performance
  * Prompting strategies like self-refinement need deeper analysis
  * Lack of real-world deployment examples

**Comments Suggestions And Typos:**

- The DoC could also be automated in other ways than by having GPT-4o act as a judge.

- Include more examples like figure 1 in the main text, not only in appendix

- Include more examples in which LLM behavior was changed by manipulation of racial attributes, unintentionally (to show ethical risks)

**Ethical Concerns:**

The paper does a good job of managing the ethics, but simulating personas along the lines of race or low performance is risky if not done with caution (reinforcing stereotypes).
Potential concern: Race-based simulation, even if not the intent of the authors, can feed into implicit bias in downstream applications.
Suggestion: Add more explicit guidelines on the appropriate and responsible use, especially for eduse applications.

**Paper Summary:**

This paper investigates whether Large Language Models (LLMs) can simulate counterfactual personas, particularly low-performing student personas in math reasoning, which is contrary to their default optimization toward accuracy. The authors propose the term “counterfactual instruction following” to define this ability. They assess multiple LLMs (e.g., GPT-4o, Claude 3.5, OpenAI o1, LLaMA 3.1) across various prompting strategies (zero-shot, one-shot, self-refinement, and two-stage prompting), using GSM8k math problems. They also test intersectional simulations combining performance level with racial identity (African American, White American, and Hispanic).

**Relevance:**

4

**Summary Of Strengths:**

- Introduces a new evaluation task and metric (DoC) that goes beyond accuracy.
- Investigates both performance and identity-level persona control.
- Includes intersectionality and shows racial attribute effects.
- Evaluates multiple prompting methods: zero-shot, one-shot, self-refinement, and two-stage.
- Human evaluation confirms DoC reliability.
- Raises critical ethical implications related to fairness and LLM bias.

**Summary Of Weaknesses:**

- Some LLMs (example OpenAI o1) can have no change in accuracy and a high DoC score - is there a way to better balance accuracy with reasoning contrast?

- Prompting techniques (primarily the two-stage methods) actually lower the performance - would be interesting to see different methods for improving them.

- Slight lack of real-world implementation (ie. putting in an education app to show the potential)

---

### Official Review · Reviewer_kfgY · 2025-07-18
**Studying simulation performance of current LLMs on synthetic personas instructed to perform badly "counter-factually", and also incorporating racial biases**

**Rating:** 6
**Overall Assessment:** 3
**Confidence:** 4

**Review:**

The prose of the paper is clear, and the work tackles an important issue in the miscalibration of human-preference fine-tuned general purpose models when performing persona simulation. Models tend to focus on getting the right answers, even when explicitly instructed to perform poorly. This tendency and arguable mis-alignment could block many usecases for simulation with generative AI, so is important to research and evaluate.

However, the takeaways from the paper are not very clear, especially when the racial bias experiments are included. The related work section is focused on persona simulation, not on the extensive literature on model bias for race, gender, etc...
The discussion focuses on the DoC metric and not the difference in accuracy between high- and low-performing personas, which is arguably at least as important as DoC scoring.

Also as stated above, I think adding the race profile information to the experiments is not well-contextualized or motivated in this work, and it adds a set of ethical questions that, although addressed in the Ethics Statement, are unnecessary for the main scientific contribution of the work, so I think this set of experiments could be removed and put into work that is focused on implicit bias evaluation in AI models, instead of counterfactual persona simulation.

**Comments Suggestions And Typos:**

Cluade --> Claude in Figure 3

**Ethical Concerns:**

Ethical concerns are listed throughout my review, and the authors also acknowledge them.
I do not think there is a major ethical issue with this work, but if the authors wish to focus on bias evaluation they should frame the work as such, instead of framing the work as counterfactual persona evaluation, but also including the racial profile experiments as a separate axis of evaluation, with minimal coverage in the introduction or background work.

**Paper Summary:**

The authors study the performance of current LLMs on tasks that require modelling personas that perform poorly at a given tasks.
Experiments focus upon measuring Degree of Contrast (DoC) between high and low-performing persona settings, with higher contrast framed as a desirable characteristic of a model that is able to accurately simulate different persona types. The DoC score is produced by an LLM judge, instructed to produce a score indicating the degree of contrast in the range 1 (no contrast) to 3 (high contrast). The authors also perform a human evaluation on the judgement task and find high correlation between llm and human judgements of contrast. LLMs are evaluated in zero shot, one shot, and self refinement settings. Experiments show that generally weaker models such as Llama3.1 are better at modelling counterfactual personals than SOTA models such as OpenAI-o1, because weaker models acheive both higher DoC scores, and much larger contrasts in tasks accuracy between high and low-performing personas.

The authors also include a separate axis of variation, focused upon measuring the effect of adding race information to persona prompts. In my view, this setup detracts from the main messaging of the paper, because it is not focused upon measuring counterfactual personas, and instead is focused on evaluating intrinsic biases in the models, which is not part of the primary narrative of the work. The addition of racial profile simulation, while relevant to some lines of work, is unnecessary in the context of this research, and greatly complicates the ethical issues associated with the research, while adding minimal scientific value.

**Relevance:**

5

**Summary Of Strengths:**

- clearly written paper well targeted to the social simulation workshop at COLM
- reasonable evaluation setup with good coverage of recent models
- important experimentally-validated observations on misalignment of current models for persona simulation

**Summary Of Weaknesses:**

- unclear messaging and evaluation wrt counterfactual personas vs racial bias axes
- insufficient contextualization of the bias experiments within current related work
- insufficient discussion of accuracy tradeoffs observed in weak / strong persona cohorts, and relationship to DoC metric (especially when DoC metric is high, but task accuracy is similar between weak / strong )

---

### Meta-Review · Area_Chair_quzC · 2025-07-21

**Recommendation:** Accept

**Metareview:**

--